# Development of Alginate-Chitosan Bioactive Films Containing Essential Oils for Use in Food Packaging

**DOI:** 10.3390/foods14020256

**Published:** 2025-01-15

**Authors:** Carla Guzmán-Pincheira, Arash Moeini, Patricia E. Oliveira, Diana Abril, Yeni A. Paredes-Padilla, Sergio Benavides-Valenzuela

**Affiliations:** 1Escuela de Nutrición y Dietética, Facultad de Ciencias para el Cuidado de la Salud, Universidad San Sebastián, Concepción 4030000, Chile; carla.guzman@uss.cl (C.G.-P.); yeniparedesp6@gmail.com (Y.A.P.-P.); 2Research Group of Fluid Dynamics, Brewing and Beverage Technology, TUM School of Life Sciences, Technical University of Munich, 85354 Freising, Germany; arash.moeini@tum.de; 3Departamento de Ingeniería de Procesos Industriales, Núcleo de Investigación en Bioproductos y Materiales Avanzados, Universidad Católica de Temuco, Temuco 4810399, Chile; poliveira@uct.cl; 4Departamento de Biología y Química, Facultad de Ciencias Básicas, Universidad Católica del Maule, Talca 3460000, Chile; dabril@ucm.cl

**Keywords:** edible films, bioactive films, chitosan, alginate, oregano oil, thyme oil, antibacterial properties

## Abstract

The effect on the physical, mechanical, and antibacterial properties of films composed of alginate-chitosan with the incorporation of oregano (EOO) or thyme (EOT) essential oils was evaluated. These films showed a thickness between 37.7 and 38.2 µm, with no significant differences for essential oil content. Water vapor permeability decreased from 4.03 (oil-free film) to 1.65 (g/msPa) × 10^−9^ in 3% EO. Mechanical properties reflected a reduction in tensile strength (*TS*) from 73 (oil-free films) to values between 34 and 38 MPa with 3% EO, while elongation (*E*%) increased from 4.8% to 10.4–11.8%. Regarding antibacterial capacity, as the concentration of essential oil increases, the antibacterial capacity also increases. On average, the increase from 1.0% to 3.0% of EOO increased the antimicrobial capacity against Gram-negative and Gram-positive bacteria. EOO outperformed EOT against *E. coli* and *L. monocytogenes*. In addition, films with 2–3% EOT showed a significant dark yellow color compared to the control. These results suggest that films with the addition of oregano and thyme essential oils can be promising for food packaging applications with the ability to improve food safety and increase product shelf life by achieving functional packaging characteristics.

## 1. Introduction

The development of edible films for use in food packaging has gained significant momentum in the last decade, given the search for sustainable alternatives to traditional packaging [1]. According to various investigations, these films, made from biopolymers such as polysaccharides, proteins, and other macromolecules, not only offer a physical barrier against moisture, oxygen, or microorganisms but can also incorporate bioactive compounds that prolong the shelf life of food [2]. However, despite the advances, there are important challenges that limit the massive implementation of these technologies, such as the stability of the films under different environmental conditions, the migration of bioactive agents through the biopolymeric matrix, the interaction between components that, in short, condition the technological properties of the films [3]. In this context, recent research suggests that developing edible films focused on the chemical interactions between mixtures of biopolymers and the bioactive agent could be a promising path in the most suitable design of composite materials for use in food. The integration of materials science with food technology is a key tool to overcome current limitations and achieve a more sustainable future in edible packaging [3].

Among the most studied biopolymers individually are alginate and chitosan [4,5,6,7,8,9,10]. These biopolymers are derived from marine sources and are food-grade, nontoxic, biodegradable, low cost, versatile, and edible. Each one presents unique physical and mechanical properties that make them attractive for the design of materials for food packaging [6,7].

Alginate is an anionic polysaccharide composed of a chain of molecules of the 1,4-β-d-mannuronic and α-l-guluronic type. It can be obtained from brown algae and is soluble in water [9]. The L-guluronic group can be cross-linked with polyvalent cations, such as calcium, allowing it to form a gel-like polymeric network [11]. The proportion of L-guluronic groups and the concentration of calcium ions determine the mechanical strength of the gel. This property allows the films to form good mechanical and barrier properties, as well as regulate, to a certain extent, the release of bioactive agents incorporated into the film, allowing the design of films suitable for bioactive food packaging [12].

Alginate films are characterized by their hydrophilic nature, attributed to the presence of hydroxyl and carboxyl groups within the alginate structure. This hydrophilicity influences their properties, being suitable barriers to oxygen but relatively poor for humidity [4] as it generates “swelling” phenomena due to water absorption [9]. The mechanical properties (tensile strength and elasticity) can be modulated by adjusting the alginate concentration and incorporating plasticizers such as glycerol or essential oils, among other compounds [13].

Chitosan is a natural polycationic polysaccharide, chemically composed of D-glucosamine and D-N-acetylglucosamine units linked in β-(1-4), with a predominance (>50%) of the former. Chitosan is obtained from several alternative sources, such as crustacean shells, molds, mushrooms, and sponges [14,15]. Chitosan films are obtained by solvent evaporation, and their antimicrobial properties are noted [15].

The linear structure of the chitosan chain, added to the presence of amino groups capable of forming intermolecular hydrogen bonds, allows the formation of films with a high degree of structuring, which translates into good oxygen barrier properties [16]. On the other hand, chitosan films can be easily modified by chemical or physical cross-linking, mixing them with other biopolymers, or incorporating active compounds to adapt their properties and functionality to specific applications [15]. However, chitosan films have some significant disadvantages, such as their high hydrophilicity, which can affect their structuring depending on the environmental humidity [14]. In addition, the solubility of chitosan in acidic solutions can restrict its application in some food groups. On the other hand, the mechanical properties of chitosan films, such as attractive force, are usually weak, requiring chemical cross-linking to improve their strength [16].

When combined, electrostatic interactions between the negatively charged carboxyl groups of alginate and the positively charged amino groups of chitosan lead to the formation of a polyelectrolyte complex [17,18,19,20]. This complex formation improves the mechanical strength, water vapor barrier properties, and stability of the resulting edible films. Furthermore, the synergistic effect of these biopolymers can improve the bioactivity of the film, offering potential benefits for food preservation and quality [20].

Incorporating bioactive agents into a biopolymeric film, such as essential oils, not only confers antimicrobial or antioxidant capacity to the edible film but can also interact with the biopolymeric matrix, affecting its physical and mechanical properties. Essential oils (EOs). EOs are found in various plant species, are characterized by being “aromatic”, and are often used as spices in gastronomy [21]. Some examples are oregano (*Origanum vulgare*), thyme (*Thymus vulgaris*), and rosemary (*Rosmarinus officinalis*) [22,23,24,25]. EOs from oregano and thyme have a broad antimicrobial spectrum (bacteria and molds) and could be a natural alternative to control pathogens and microorganisms that cause food spoilage [26,27,28,29,30]. Films containing EOs can be considered active packaging since they isolate food from the environment and simultaneously show an antimicrobial effect, preventing food spoilage and the proliferation of pathogens [29,30,31,32,33,34]. The antimicrobial capacity of EOs is due to the presence of volatile compounds, mainly of the phenolic type, such as thymol and carvacrol [29]. It has been demonstrated that these compounds act at the cellular level, generating lipid membrane disruptions in microorganisms, which can lead to cell lysis [35,36]. This antimicrobial capacity has been observed against various pathogens such as *E. coli* [37,38], *St. aureus* [38,39], *Salmonella* ssp. [40,41], and *Listeria monocytogenes* [41], among others [42]. Other phenolic compounds, such as rosmarinic acid [43] and eugenol [44], also exhibit significant antimicrobial effects and may synergize with essential oils. Research on these compounds allows the availability of bioactive compounds to ensure the safety and shelf life of fresh or processed foods [45].

The development of biocompatible and functional films from alginate and chitosan combined [17,18,19] with essential oils is based on the complex structural interaction mediated by ionic charges between these polymers and essential oils [46]. This synergy is crucial for forming homogeneous matrices that meet the demands of food packaging applications. However, one of the biggest challenges lies in the rapidity of the interaction between alginate and polycations [9], which can result in a heterogeneous morphology of the films obtained. Consequently, the mechanical and functional properties of the films are inconsistent, manifesting in zones of variable strength, from robust areas to more fragile regions [13]. This heterogeneity not only affects the mechanical integrity of the films but also affects their microtopography [13], which can vary from rough surfaces to smooth and agglomerated areas, showing a lack of uniformity that could positively or negatively affect the films’ performance.

The application in food packaging. On the other hand, the incorporation of essential oils represents not only an alternative to enhance the antibacterial properties of films but also a way to modulate their mechanical and physical characteristics [13,46]. Due to their complex nature and diverse chemical structure, essential oils can induce additional interactions in the polymeric network formed by alginate and chitosan, contributing to the homogeneity of the films. This work is aimed at evaluating the effect of interactions between biopolymers and essential oil and their effect on the technological properties of edible films, a key aspect in developing new materials for food packaging.

## 2. Materials and Methods

### 2.1. Materials

The essential oils used in this study were obtained by hydrodistillation from oregano (*Origanum vulgare*) and thyme (*Thymus vulgaris*) leaves (Sigma-Aldrich, St. Louis, MO, USA). The main components of both oils are carvacrol, thymol and p-thymene (EOO: 426 g/L carvacrol, 218 g/L p-thymene and 139 g/L thymol. EOT: 132 g/L of carvacrol and 332 g/L of thymol. Sodium alginate (CAS number: 9005-38-3, medium viscosity), Chitosan (CAS number: 9012-76-4; high molecular weight, and degree of deacetylation 75%), calcium chloride, acetic acid, glycerol, and Tween-80 were purchased from Sigma-Aldrich (St. Louis, MO, USA). The antibacterial capacity of thyme and oregano essential oils was evaluated on the following bacterial cultures: *Escherichia coli* (ATCC 25922), *Salmonella enteritidis* (ATCC 13076), *Staphylococcus aureus* (ATCC 6538) and *Listeria monocytogenes* (ATCC 7644). These cultures were obtained from the Food Microbiology Laboratory of the Food Engineering Department of the Universidad del Bio Bio, Chile.

### 2.2. Methods

#### 2.2.1. Determination of Antibacterial Properties of Essential Oils

The estimation of the minimum inhibitory concentration (MIC) and minimum bactericidal concentration (CMB) was carried out using dilution in broth [47]. EOs dilutions from 10.0 to 0.01% *v*/*v* were used. The bacterium culture was seeded in trypticase soy broth (TSB, GranuCult^®^, Merck, Darmstadt, Germany) and cultivated in an oven (Memmert 55 L, UN 55 model) for 24 h at 35 ± 1 °C until cell concentration was reached. of 10^6^ CFU/mL that was confirmed with the McFarland scale [25]. The MIC values were taken as the lowest concentration of oil that prevents visible bacterial growth after 24 h of incubation at 37 ± 1 °C, and MBC as the lowest concentration that completely inhibits bacterial growth. TSB with methanol and without essential oils was used as a negative control, and inoculated TBS was used as a positive control. Each experiment was performed in triplicate.

#### 2.2.2. Preparation of Alginate-Chitosan Films

For the preparation of the chitosan solution, the methodology indicated by Lijun, et al., [48] was applied, with modifications: 4 g of chitosan were dissolved in 200 mL of distilled water acidified with acetic acid (1.0% *v*/*v*) stirring for 24 h at 1000 rpm at 20 ± 1 °C. Finally, the chitosan solution was sonicated for 5 min to remove bubbles and filtered through cellulose paper (Whatman, 3 µm). The plasticizer used was glycerol (0.243 g glycerol/g chitosan). In parallel, a 1.0% *w*/*v* sodium alginate solution was prepared; the alginate dilution was acidified with 1.0% *v*/*v* acetic acid to prevent the insolubilization of the chitosan in the sodium alginate solution. The chitosan-alginate mixture was made in a 1:1 ratio in a 500 mL beaker under constant stirring at 200 rpm (Amnt EQL-00896, Numak, London, UK) for 10 min [15].

#### 2.2.3. Preparation of Alginate-Chitosan Films with Antimicrobial Agents

Previously, we performed an antibacterial evaluation on *E. coli* and *St. aureus*, with concentrations of essential oil of 0.1, 0.2, 0.3, 0.5, 1.0, 2.0, and 3.0% *w*/*v*, the last 3 being those in which an inhibitory halo was evidenced. Essential oils of oregano and thyme were added to a freshly prepared chitosan/alginate solution (1:1 ratio) under gentle stirring. Oregano oil (EOO) or thyme oil (EOT), previously mixed with Tween-80 emulsifier (0.250 g/g oil), were added to the biopolymer mixture at final concentrations of 1.0; 2.0 and 3.0% *w*/*v* respectively. The solutions were homogenized with Ultra Turrax equipment (IKA^®^ Werke, Germany) at 13,500 rpm for 3 min. Subsequently, the solutions were ultrasonicated in an ultrasonic bath (Biobase, Mod. UC-30A, Fremont, CA, USA) for 20 min to eliminate bubbles. The emulsions were transferred to Petri dishes (at a rate of 30 mL per capsule) and placed in an oven (55 L Memmert, UN 55 model, Eagle, WI, USA) to dry at 35 ± 1 °C until constant weight (48 to 62 h). After that, one milliliter of CaCl_2_ (0.5% *w*/*v*) solution was added to the film surface and left in contact for 5 min. The excess solution was removed. Petri dishes with biocomposite films were dried for 3 h at 35 ± 1 °C. Finally, the films were removed from the Petri dishes and placed in a desiccator at 20 °C with 30% humidity until use.

#### 2.2.4. Determination of Physical Properties (Methodology)

##### Film Thickness

The film thicknesses were measured with a digital micrometer (Electronic Outside Micrometer, Semmes, AL, USA) in different areas of the film. The thickness was considered as the average of 5 measurements [13].

##### Color, Yellowness Index, and Whiteness Index

The film’s color was measured with a CR-300 colorimeter (Minolta Camera Co., Ltd., Osaka, Japan) through a calibrated standard plate (a = 0.20; b = 1.84; L = 97.14). The CIELab color space was used, and lightness (*L**), and color parameters *a** (red-green) and *b** (yellow-blue) were measured using a D65/10° illuminant observer. Total color difference (ΔE) was defined between the alginate-chitosan control film (*L**_0_, *a**_0_, *b**_0_) and the alginate-chitosan films with EOs (*L**, *a**, *b**), where the values *L** (black 0 to white 100), ^a*^ (red 120 to green −120), and *b** (yellow 120 to blue −120) correspond to whiteness, redness, and yellowness, respectively (Equation (1)). Furthermore, the yellowness index (*YI*) and whiteness index (*WI*) were calculated using Equations (2) and (3), respectively [18].

(1)
$$∆E=\sqrt{{\left(L_{0}^{*}-L^{*}\right)}^{2}+{\left(a_{0}^{*}-a^{*}\right)}^{2}+{\left(b_{0}^{*}-b^{*}\right)}^{2}}$$



(2)
$$YI=142.86\;(b^{*}/L^{*})$$



(3)
$$WI=100-\sqrt{{\left(100-L^{*}\right)}^{2}+{\left(a^{*}\right)}^{2}+{\left(b^{*}\right)}^{2}}$$


##### Water Vapor Permeability (*WVP*)

Water vapor permeability (*WVP*) was determined using the modified ASTM E96-95 gravimetric method proposed for hydrophilic films [49]. Film discs were mounted in permeability dishes (Fisher Scientific, San Luis, MO, USA), to which 10 mL of distilled water was previously added. The dishes were placed in a desiccator containing saturated sodium chloride solution (75% relative humidity) to generate a 75/100% relative humidity gradient between both sides of the film. The air around the film was homogenized using a small fan inside the desiccator. Finally, the system was placed in an air circulation oven at 25 ± 1 °C. The weight of the capsules was checked every 2 h for a period of 24 h. The *WVP* was estimated using the regression analysis of the Equation (4):
(4)
$$WVP=\left(\frac{w}{t}\right)\times \left(\frac{Tks}{A\;P^{sat}\times ({RH}_{in}-{RH}_{out})}\right)$$

where *Tks* is the average thickness of the edible films, A is the permeation area (5.5 × 10^−3^ m^2^). *P^sat^* is the saturation pressure of the atmosphere (25 °C), *RH_in_* is the relative humidity inside the dish and *RH_out_* is the relative humidity of the atmosphere. The *w*/*t* term was calculated by linear regression from the points of weight and time increase in the constant velocity period [30]. Three replicates were obtained for each sample.

#### 2.2.5. Determination of Mechanical Properties

The tensile strength (*TS*) and the percentage elongation to break (%*E*) of the films were measured in a texture analyzer (TA-XT2, TA Instruments, Surrey UK) according to the standardized method ASTM D882-95 (ASTM nineteen ninety-five). The films were cut into 20 by 90 mm strips and conditioned at 25 °C and 75% relative humidity in a desiccator with saturated Mg(NO_3_)_2_ solution for 72 h. A set of jaws with a separation of 50 mm and an operating speed of 1 mm/s were used for mechanical properties measurement [13]. The *TS* was read directly from the equipment in units of MPa, and %*E* was calculated by Equation (5):
(5)
$$\%E=\left(\frac{L}{L_{0}}\right)\times 100$$

where *L* is the length at the breaking point (mm), and *L*_0_ is the original length (mm). All measurements were performed in triplicate.

#### 2.2.6. Scanning Electron Microscopy (SEM)

The microstructures of the microtopographic and cross-sect films were obtained and examined using a scanning electron microscope (SEM, JEOL JSM-6380LV, Tokyo, Japan). Previously, the films were frozen with liquid nitrogen and fractured into approximately 5 × 3 mm pieces. The pieces were fixed in a sample holder and coated with a nanometric layer of gold using sputtering equipment (Aname, SC7620. Madrid, Spain). The samples were observed using an accelerating voltage of 15 kV and 3000× of magnification.

#### 2.2.7. Thermogravimetry Analysis

For the TGA evaluation of the samples, the methodology described by Jinman et al. [50] was used: The measurements were carried out in a thermogravimetric analyzer (Netzsch TG 209-F3, Waldkraiburg, Germany), and the temperature increase was from 30 to 600 °C at a rate of 10 °C/min.

#### 2.2.8. FTIR

The samples were dried for 12 h at 60 °C. The dried samples were subjected to FTIR absorption spectrum analysis using an Alpha T FTIR spectrometer (Bruker, Mannheim, Germany) equipped with an attenuated total reflectance (ATR) unit. Scans were obtained over a spectral range of 4000–500 cm^−1^ with a resolution of 4 cm^−1^ for each sample. The scans were processed using Origin Pro software (1.5 2024). To enhance the visualization and comparability of the FTIR spectra, Min-Max normalization was applied, scaling the absorbance values to a range of 0 to 1 [50].

#### 2.2.9. Antibacterial Effect of the Alginate-Chitosan Films

The agar diffusion method was used to determine the antibacterial effect according to the method proposed by Chen et al. [51]. Sterile 21 mm diameter film discs were placed on Muller Hinton Agar (MHA, GranuCult^®^ Merck, Darmstadt, Germany), previously inoculated with 0.1 mL of trypticase soy broth (TSB, GranuCult^®^, Merck, Darmstadt, Germany) 10^5^–10^6^ CFU/mL bacterial strains. The Petri dishes (90 mm) were incubated at 37 ± 1 °C for 24 h. Finally, the diameter of inhibition generated from the edge of bacterial growth around the film disc was measured. These tests were performed in triplicate.

### 2.3. Statistical Analysis

Statistical analyses were performed using IBM SPSS version 21 software. The effects of each variable on the functional and antibacterial properties of the films were analyzed using a one-way ANOVA test. Differences between the mean values of film properties were determined by Dunnett’s T3 least significant difference test for multiple comparisons. Data are represented as mean ± standard deviation. Statistical significance was considered at *p* < 0.05. All analyses were performed in triplicate, except for color, thickness and mechanical properties (*TS* and %*E*), which were performed in quintuplicate.

## 3. Results and Discussion

### 3.1. Determination of the Minimum Inhibitory Concentration (MIC) and Minimum Bactericidal Concentration (MBC) of Essential Oils

EOs have generally shown broad antimicrobial activity, particularly against spoilage and pathogenic bacteria [23]. Previous studies have observed a higher concentration of essential oils of thyme and oregano in edible films for meat, resulting in significant antibacterial activity [22]. Similarly, EOT and EOO enhanced the helpful life of the Trout [52]. To evaluate the antimicrobial capacity of essential oils, it is necessary to determine the minimum inhibitory concentration (MIC) and the minimum bactericidal concentration (MBC). Both parameters previously indicate the antibacterial power of each oil. The results of this stage are shown in Table 1. The results showed that Gram-positive bacteria (*S. aureus* and *L. monocytogenes*) were more vulnerable to essential oils, requiring low concentrations to cause an inhibitory or bactericidal effect. On the other hand, Table 1 also shows that the EOO generally showed a greater antibacterial capacity than EOT. The antimicrobial capacity of essential oils depends on their active compounds, predominantly monoterpenes (thymol, carvacrol, among others). The antibacterial mechanism of action of these active compounds is based on the alteration of the cytoplasmic membrane through proton interactions and electron flow, which alters the transport mechanisms at the cell membrane level and can generate cytoplasmic coagulation effects [36]. Therefore, the lower susceptibility observed in Gram-negative bacteria (*E. coli* and *S. enterica*) could be explained by the more complex structure of these bacteria due to an additional outer membrane [36]. It is important to note that EOO showed a significantly higher inhibitory power (*p* < 0.05) than EOT against the same microorganisms. On the other hand, the concentrations of oil used to achieve bacterial lysis were lower, demonstrating that EOO has greater bactericidal power. These results approved previously reported observations [26]. However, variations in values are probably because of environmental conditions, the types of active ingredients, the geographic origin of the plant, and the extraction methods.

### 3.2. Physical Properties

Figure 1 shows, as an example, some films obtained after the dehydration process. The yellow color increases as the essential oil content increases. To the touch, the films with EO were more flexible than the control (0.0% EO).

#### 3.2.1. Film Thickness

The thickness of the films ranged from 37.7 to 38.2 µm (Table 2). The thicknesses of all films formulated with EO (regardless of type) increased compared to the control film because the incorporation of EO in the films decreases the emulsion density. Consequently, the same volume has less solids than a solution without EO [13,53]. However, this increase in film thickness was not statistically significant. The mass and volume variations generated in filmogenic matrices depend mainly on the constituent components, their properties (density, for example), and the interactions between them in terms of electrostatic repulsions or attractions, hydrogen bond formation or others, which can result in more cohesive or looser structures. On the other hand, the molecular structure of alginate contains -COO- groups, which under acidic conditions are converted into -COOH, something similar occurs when interacting with chitosan. This contracts the molecular chain, reducing hydrophobicity. Due to the aforementioned reason, the incorporation of a particular agent in a filmogenic matrix, as is the case of EO in alginate/chitosan matrices, could increase, decrease or maintain the same thickness when dosed at a constant mass.

#### 3.2.2. Color Measurement, Yellowness Index (*YI*), and Whiteness Index (*WI*)

The color tone of the film can affect consumer acceptance of the product [33]. That is why the evaluation of the color of a film is a crucial requirement, particularly when structural components influence this parameter. Essential oils often give a yellow hue to the films. However, this effect depends on the type and concentration of the essential oils [54]. Table 2 shows the effect of incorporating EOs (oregano or thyme) on the color parameters. The result showed that increasing the EOs content from 1 to 3% increased ΔE and reduced the lightness (L* value) from 98 in control films to 86 for 3% oregano-formulated films. In the case of EOT, similar reductions were obtained from 98.5 (control) to 86.3 (*p* < 0.05), a result that agrees with previous research where the lightness value tends to decrease when emulsions are prepared filmogenic based on essential oils [13,39]. Our results indicated that both oregano and thyme essential oils significantly reduced the luminosity of the films. However, these differences became evident as the EOs concentration in the film increased. On average, the blank films’ *L* value was 98.5, while the films containing 1% essential oils were 87.8 and 88.3 for oregano and thyme, respectively. This effect is due to the preparation of the emulsion before the formation of the film since the essential oils are arranged as a dispersed phase in the matrices, which affects the refractive index of light and increases its dispersion [40]. On the other hand, it was possible to show a significant variation in the color parameter a* (red-green). In this sense, the blank film gave a value of −1.5, while the film with EOO varied between −3.5 to −3.6 and thyme between −3.3 to −3.7. It was also possible to show a slight displacement of the color towards the greener zone of the CIELAB colour space. Regarding the parameter b* (yellow-blue), it was observed that, regardless of the type of EOs, the presence and increase in the concentration of these compounds resulted in a shift in colour towards the yellowest area of the CIELAB space. Besides, the blank film gave a b* value of 5.5, while for alginate/chitosan-EOO films, the b* value varied between 11.6 and 11.7, and for alginate/chitosan-EOT films, between 11.6 and 12.0. Comparing the blank film, *L*, a*, and b* parameters have been varied in the EOs incorporated films, especially in the b* parameter, where significant statistical differences were evident. Consistent with this, the total color variation Δ*E*, including the effect of all parameters (Equation (1)), also showed significant differences between the blank (4.3) and the EOs formulated films (12.1 and 14.0), mainly due to the change in the b*. Finally, the yellowness index (YI) indicated a significant change between 7.94 for the blank film and around 20, regardless of the type of oils for formulated films, confirming the yellower color of incorporated films. Sharma et al. [33], who worked with thyme and clove essential oils, also observed the effect of incorporating EOs on the variation of the b* parameter. They evidenced the change from 1.3 to 1.9 by increasing the thyme essential oil from 0 to 10% in PLA/PBAT films. This effect was more noticeable in clove essential oil, reaching 7.78 [33]. Regarding the whiteness index (*WI*), a significant decrease could be due to the incorporation of essential oils that led to a darkening of the film. For the same reason, the change in L could result from increased essential oils in the filmogenic emulsion.

#### 3.2.3. Water Vapor Permeability

Food packaging films require good barrier capabilities, particularly with ambient humidity. Limiting water migration between the medium and the feed maintains effective control over unwanted hydration in the feed, which can increase water activity and promote microbial growth. Consequently, the film must cancel or minimize moisture transfer between the environment and the food surface [55]. For this reason, evaluating water vapor permeability is necessary for food packaging. As shown in Table 2, regardless of the type of oils, the increase in EOs content significantly decreases water vapor permeability (*p* < 0.05). It can be due to the EOs’ hydrophobicity and can confirm the homogeneous and even distribution of EOs as a dispersed phase throughout the films. In several studies, the water vapor barrier (*WVP*) effect has been reported in biopolymeric films that incorporate essential oils in their formulation.

#### 3.2.4. Determination of Mechanical Properties (Tensile Strength and Elongation at Break)

The tensile strength (*TS*) and the elongation at break (%*E*) represent the mechanical properties of the films, which depend on their microstructural characteristics. These variables are crucial to define the behavior of the polymeric matrices in terms of plasticity and resistance of the material, which ultimately determines the quality of the packaging material. The results of this analysis are shown in Figure 2 (*TS*) and Figure 3 (%*E*). The results indicated that the incorporation of essential oils (oregano or thyme) in concentrations of 1, 2, or 3% significantly decreased the *TS* (*p* < 0.05).

On the other hand, the flexibility of the films is represented by the elongation at break (%*E*). It was observed that the incorporation of essential oils (oregano or thyme) generated a significant increase in *E*% compared to the control without essential oils (*p* < 0.05). The increase in the concentration of oregano or thyme oil from 0 (control) at 2.0% increased *E*% from 4.12% to 12.50% on average (Figure 3), which can be explained as a plasticizing effect of the film by the essential oils. Indeed, incorporating essential oils in the films can weaken the internal polymer bonds resulting from replacing polymer-polymer interaction with polymers-EOs interaction through hydrogen bonds and hydrophilic-hydrophobic interactions [7]. For this reason, EOs-incorporated films showed an increase in elongation and flexibility, leading to more fragile films with excellent stretchability [13,33]. However, a 3.0% concentration of essential oils (oregano or thyme) makes the film less elastic by over-plasticizing the film [13].

A similar result was reported by Karami et al. [11], where the effect of yarrow essential oil (EOY) in gelatine/sodium alginate films was investigated. The result showed that the increasing EOY from 0 to 3% decreased *TS* by 59.8%, while *E*% increased by 60.8% [45]. In addition, Zhou et al. [32] worked with starch films and the incorporation of cinnamon essential oil and concluded that an increase in essential oil from 0 to 2.5% managed to reduce *TS* by 43.7%, while the E% increased by 99.1%. In this work, when the essential oils content was increased from 0 to 3% *w*/*v*, the *TS* decreased to 47.1 and 44.6%, while the *E*% increased to 132 and 145% in EOO and EOT, respectively.

#### 3.2.5. Microstructural Features

SEM micrographs show the formation of microstructures generated by the biopolymers (alginate-chitosan) and the incorporation of essential oil. Figure 3 shows the surface and cross-sectional area of the films obtained. No microstructural differences were observed associated with the type of oil (oregano or thyme), but with the concentration of oil. Figure 4 and Figure 5a,d shows the surface and cross-sectional microstructures of the control film (without essential oil). In this regard, a smooth film can be observed, without pores or cracks [56], the cross section (Figure 4d and Figure 5d) is smooth and compact, without cross-linked structures. The incorporation of oregano or thyme essential oils generated a heterogeneous surface with gelled areas that increased proportionally as a function of the m essential oil concentration. The arrangement of the alginate molecules and the arrangement of the emulsified oil during film formation may result in some concave or convex irregularities on the surface [13,56]. The cross-section of the films showed a spongiform cross-linked structure. The pores of the film increased with the concentration of the essential oil, probably due to the mobilization of the oil from the inner structure to the surface due to the temperature during drying, which may affect the microstructure [25].

#### 3.2.6. Thermogravimetry Analysis

Figure 6 shows that all samples have various thermal decomposition processes. It is known that the first thermal effect occurring at a maximum decomposition rate (Tpeak) below 100 °C is due to the evaporation of free and bound water from the samples [57]. In this process, films shown an associated mass losses ranging from 5.1–10.6%. Some studies point out that the method used for chitosan-alginate polyelectrolyte complex drying could influence their supramolecular structure, and hence the amount of water inside them once dried [58].

The thermogram of chitosan (Figure 6A) indicates a second thermal degradation effect within the range of 251–402 °C, with Tpeak at 310 °C, and associated mass loss of 50.4%. This process is associated with the first stage of pyrolysis of polysaccharides, where simultaneous sugar ring dehydration, chain depolymerization, and decomposition of the acetylated and deacetylated units of chitosan occur [21,57]. At higher temperatures a continuos polymer degradation occurs, which is due to residual polysaccharide decomposition. This behavior is similar to previously observed during chitosan thermal degradation by other authors [57].

On the other side, the TG-DTG curve of alginate (Figure 6B) shows four thermal effects. The second stage shows the major polymer degradation and occurs withing a temperatura range 191–266 °C, with Tpeak 244 °C and a mass loss 24%. It is due to the polymeric chain hydrolysis, decarboxylation, decarbonilation and dehydration of the sugar ring [48]. Another two-process shown a maximum decomposition rate at 284 °C and 399 °C, with an associated mass losses around 11% each. These processes must be associated with polymer pyrolysis and residual decomposition of the polysaccharide, respectively.

In calcium-crosslinked chitosan-alginate polyelectrolyte complex (CHI-ALG) films, thermogravimetric curves (Figure 6) reveal two thermal effects with maximum decomposition temperatures at 216 °C and 265 °C, respectively. The weight losses associated with the second decomposition effect (42.4%) is 1.7-times higher than the first one (24.6%). It is worth noting that both Tpeak temperatures are lower than the maximum decomposition shown by neat polysaccharides (Table 3, Figure 6A,B). The above results suggested that the polyelectrolyte complex formation leads to a decrease in the thermal stability of the obtained film. This could be due to the catalytic effect of Ca^2+^ ions on thermal decomposition of the complex, like previously observed with Ba^2+^ ion [57]. In this complex, residual decomposition begins above 500 °C, indicating the formation of a more complex and thermally stable structure than in single polysaccharides. In previous work, it was reported the composition of the chitosan–alginate polyelectrolyte complex is independent of the alginate chemical composition as well as the chitosan molecular weight [59]. Additionally, the degree of complexation between them had been found to be 0.51 regardless of the sodium alginate composition. In this sense, the study of thermal stability of CHI-ALG complex with different polymer ratio shown similar shape than TG-DTG curves from present work [57]. Moreover, these authors reported a first Tpeak near 180 °C and a second Tpeak in the range 218–275 °C, depending on the chitosan content. It is worth noting that the second temperature effect of the CHI-ALG films in this research occurs within the temperature range previously reported.

Finally, a comparable thermal breakdown pattern is seen between the samples containing essential oils of thyme and oregano. In contrast to the CHI-ALG crosslinked films, these samples exhibit five thermal degradation steps. The initial effect in each group of oil-containing films occurs within temperature ranges of 179–240 °C and 204–241 °C, respectively, with Tpeak approximately at 220 °C. The recorded mass losses varied with the oil concentration. This effect in EO containing films start at higher temperatures than in CHI-ALG sample, suggesting an increased thermal stability of the bioactive films. It is probably due to the new interaction (e.g., hydrogen bridges) formed between oil components and polymeric matrix providing higher chemical stability. In this stage, the simultaneous breakdown of the volatile components of the essential oil and the first step of the polyelectrolyte complex decomposition may occur. The third and largest thermal effect of these films exhibited a Tpeak ranging from 260 to 274 °C, with associated mass losses of 24.8% to 34.4%. This result resembles the third decomposition step of CHI-ALG films (Tpeak 265 °C) but exhibits reduced mass losses. A fourth and five thermal degradation stage in oil-loaded films revealed Tpeak at 400 and 476 °C, respectively. In the fourth degradation step, it can be observed that going from EOT1 to EOT3 the mass losses vary from 14.7 to 27.5% and from EOO (1%) to EOO (3%) it change from 15.5 to 26.8%, respectively. These results are quite similar and are independent of the type of EO included in the polymeric matrix. It suggests the addition of EO change the thermal decomposition behavior and stability of CHI-ALG interpolymeric complex. It is possible that both effects could be due to the residual decomposition of the mixture, which is generated by novel compounds. Such products were produced through a reaction between the polymeric matrix and the chemical constituents of the essential oil. Films with analogous polymeric compositions, loaded with aromatic compounds, have exhibited similar shapes [46].

#### 3.2.7. FTIR

Fourier-transform infrared spectroscopy with attenuated total reflectance (FTIR-ATR) was performed on each sample, including the blank. Essential oils are complex mixtures of various components, and their IR spectra represent an overlap of the spectra of individual compounds with a combination of multiple vibrational modes [47]. The obtained spectra are presented in Figure 7.

In Region I, all spectra exhibit an absorption band peaking between 3500 and 3300 cm^−1^, corresponding to overlapping the symmetric and asymmetric stretching of -OH groups from chitosan and alginate chains [60]. Similarly, the hydroxyl groups of the polyphenols from the essential oils form an extensive hydrogen-bonding network with the polysaccharide chains, as evidenced by the broadening of the -OH absorption band. Additionally, this absorption band also includes the symmetric and asymmetric stretching of -NH groups in chitosan [61]. Region II shows an absorption band at 2973 cm^−1^ which belongs to the vibration of the -CH_2_ groups from both polymers. Also, the band at 2865 cm^−1^ is attributed to the symmetric stretching of C-H bonds in methyl groups (-CH_3_) of chitosan. The last signal is associated with secondary amides present in the chitosan polymer chain, likely due to the presence of acetylated units [62]. Once mixed the biopolymer to form the CHI-ALG film, the absorption band from 2865 cm^−1^ shifts to 2886 cm^−1^, which can be attributed to new hydrogen bond interactions formed between the acetylated amines of chitosan and the -OH and -COOH groups of alginate [62].

**Region I and II.** In films containing essential oil, new absorption bands appear in the 2900–2930 cm^−1^ region, corresponding to the stretching vibrations of C-H bonds from methylene groups. These indicate the presence of terpenes and other organic compounds in the essential oils. The methylene band overlaps with the methyl absorption band in the spectra of films loaded with thyme oil [63].

**Region III and IV.** In the chitosan spectrum, the so-called amide bands appear at 1657 cm^−1^ (amide I), 1560 cm^−1^ (amide II), and 1319 cm^−1^ (amide III), respectively. These bands have been assigned to the valence vibration of the carbonyl group (ν_C=O_), the bending δ_NH_, which is attributed to the NH deformation in the plane of the amide bond, and the valence vibration of the C-N bond (ν_C-N_), respectively. In alginate spectra, the absorption peak due to the antisymmetric valence vibration of carboxylate ions ν^as^_COO-_ appears at 1590 cm^−1^, and the band located at 1419 cm^−1^ is assigned to its symmetric valence vibration (ν^s^_COO-_). The CHI-ALG film spectrum differs from that of neat polysaccharides because it shows an overlap of the characteristic absorption bands of CHI and ALG. The only difference observed is the broad band at 1560 cm^−1^ with higher intensity [64], which corresponds to the symmetric bending δ^S^_NH3_+ characteristic of protonated amino groups. This is evidence of the presence of interchain salt bridges between CHI-ALG forming a polyelectrolyte complex.

**Region V and VI.** In the absorption spectra of films with 2% and 3% oregano oil, a characteristic band is observed around 810 cm^−1^, attributed to the out-of-plane bending of C-H bonds in the aromatic ring of carvacrol. Additionally, the band at 865 cm^−1^ corresponds to the aromatic skeletal vibration of carvacrol [65]. The in-plane deformation vibration of the =CH_2_ group, associated with the alkenes present in the essential oils, is recorded near 1410 cm^−1^. Furthermore, representative bands in the 930–945 cm^−1^ region are linked to the presence of γ-terpinene, a key precursor in the biosynthesis of carvacrol and thymol [63].

#### 3.2.8. Antibacterial Effect of Alginate-Chitosan Films

The antibacterial effect of incorporated EOs within the biopolymer matrices depends on migrating the EOs to the film’s surface. Figure 4c and Figure 5c show granular structures with increasing essential oil content in the formulation. These granularities also increase the surface area of the films and may facilitate the migration of the essential oils through the film. As mentioned in the SEM section, the porosity is directly influenced by the concentration of EOs. Consequently, the antibacterial capacity of the films depends not only on the oil concentration but also on the surface migration capacity. Table 4 shows the antibacterial results of the inhibition halos generated by the films. As the oil concentration in the film increases, the antibacterial capacity also increases, demonstrated by the generation of the inhibition halo (Figure 8). On average, the increase from 1% to 3% of EOO increased the antimicrobial capacity against Gram-negative bacteria from 24 to 37 mm, while for EOT, it was from 28 to 39 mm. In the case of Gram-positive bacteria, 1 to 3% of oregano oil increased the inhibition halos between 26 to 46 mm, while for EOT, it was from 34 to 55 mm. This situation had already been verified when testing the antibacterial efficacy of the pure oils, which is consistent with the results observed in previous research [37].

The essential oils (oregano or thyme) incorporated in films showed an interesting antibacterial effect that was more intense in higher concentrations of EOs. The alginate/chitosan-EOT film was particularly effective against *L. monocytogenes*, showing significantly larger inhibition halos than other bacterial strains. In most cases, the inhibition halos against *E. coli*, *S. aureus*, and *L. monocytogenes* were considerably lower in alginate/chitosan-EOO compared to alginate/chitosan-EOT films with the same concentration.

## 4. Conclusions

Research on the development of bioactive films based on alginate and chitosan, enriched with essential oils of oregano or thyme, offers an alternative to developing bioactive materials for food packaging. These films are suitable for wrapping various food products, from fresh meats to dairy products such as cheeses. The addition of essential oils not only provides significant antimicrobial properties but also conditions their technological properties. In this context, the interaction between alginate, chitosan, and essential oils significantly influenced the physical, mechanical, and antibacterial properties of the films. No significant differences were detected between the essential oil content and the thicknesses of the films obtained, the latter being between 38.2 and 37.7 µm. Regarding the CIELAB color parameters, only the films with EOT between 2.0 and 3.0% showed significant differences with respect to the *L**, a*, and b* parameters, which was due to a greater orientation to the dark yellow color, concerning the control and the films with EOO. Water vapor permeability was significantly affected by the essential oil concentration. Water vapor permeability was inversely proportional to the EOs concentration in the films. A 3% EOs concentration reduced the permeability by 1.80 to 1.65 (g/msPa) × 10^−9^, compared to the control film (without EOs) of 4.03 (g/msPa) × 10^−9^. In terms of mechanical properties (*TS* and *E*%), it was shown that incorporating EOs significantly decreased the *TS* while increasing the *E*%. The control film (without EO) had a *TS* of 73 MPa and an *E*% of 4.8, while the films with 3% EO gave *TS* between 38 and 34 MPa and *E*% between 10.4 and 11.8%. The incorporation of EO into the films significantly enhanced the antimicrobial capacity. However, significant differences were identified between EOO and EOT at 3% for the species *E. coli* and *L. monocytogenes*, where EOO was more effective than EOT. In future work, we expect to evaluate the films on salmon and chicken fillets, as well as fresh and aged cheeses, to assess the effect on the shelf life of the food.

## Figures and Tables

**Figure 1 foods-14-00256-f001:**
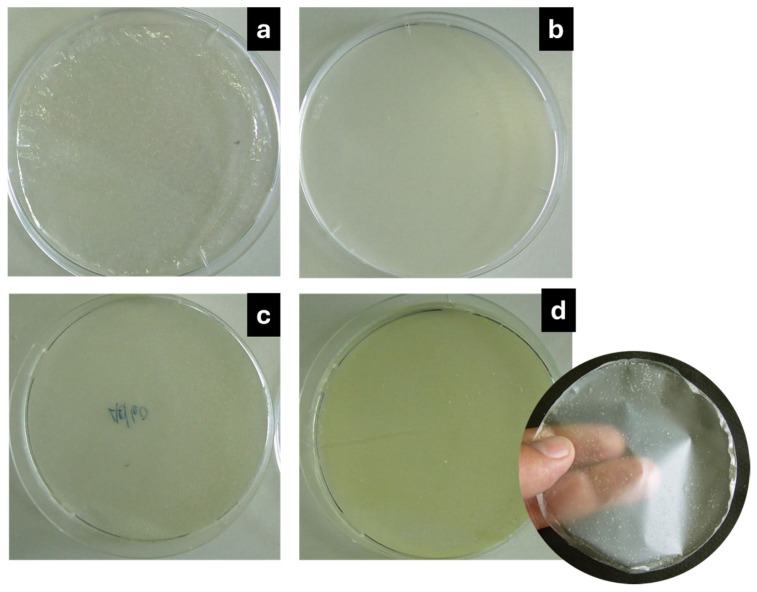
Film samples obtained (**a**) Control (0% EO), (**b**) 1.0% EO, (**c**) 2.0% EO, and (**d**) 3.0% EO.

**Figure 2 foods-14-00256-f002:**
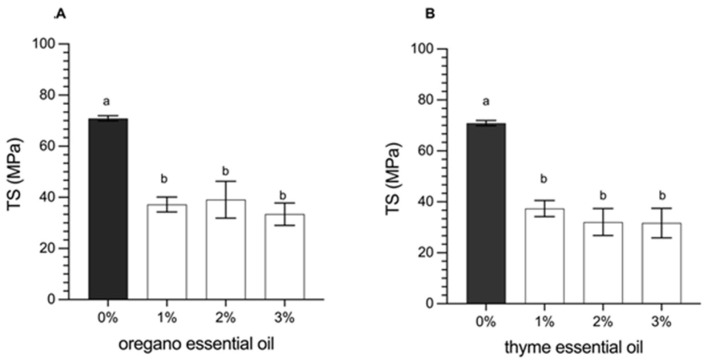
Tensile strength (*TS*) of the film with (**A**) oregano oil and (**B**) thyme oil. Data are reported as mean ± standard deviation (*n* = 5). Mean values with different letters indicate a significant difference (*p* < 0.05).

**Figure 3 foods-14-00256-f003:**
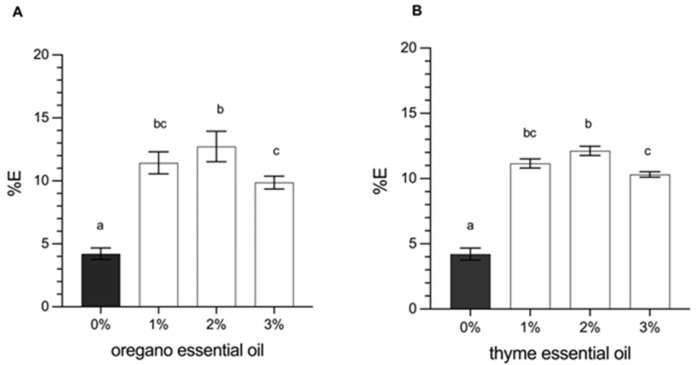
Elongation at break (%*E*) of the film with (**A**) oregano oil and (**B**) thyme oil. Data are reported as mean ± standard deviation (*n* = 5). Mean values with different letters indicate a significant difference (*p* < 0.05).

**Figure 4 foods-14-00256-f004:**
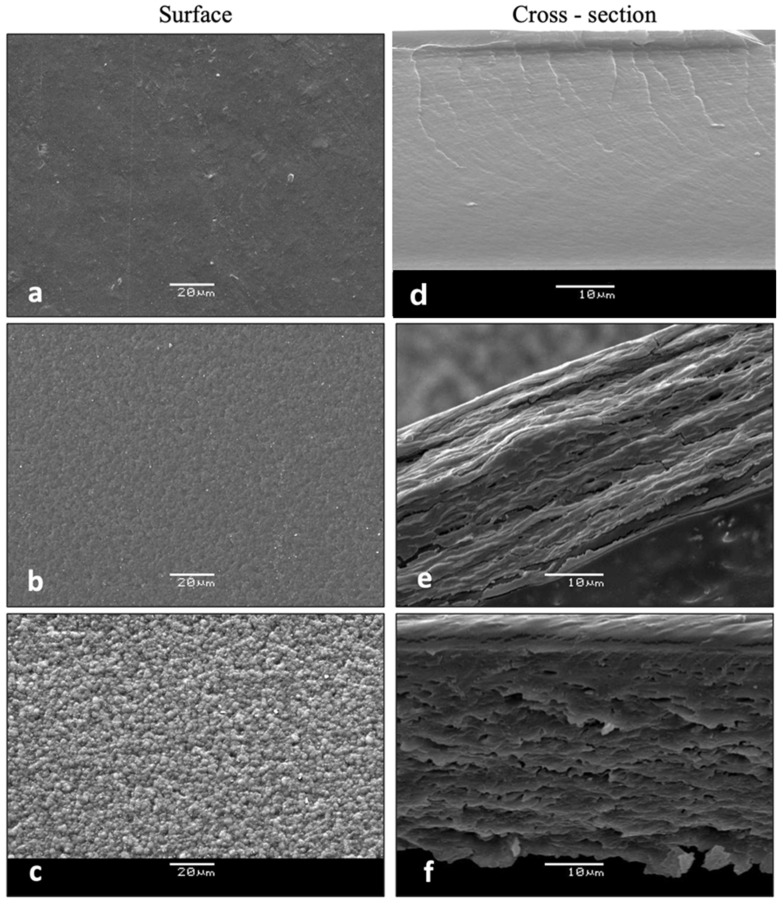
SEM micrographs showing the surface structure of (**a**) control alginate-chitosan film, and with the addition of (**b**) EOO 2%, (**c**) EOO 3%, and cross-section of (**d**) control alginate-chitosan film and with addition of (**e**) EOO 2% (**f**) EOO 3%. Oil (magnification 3000×).

**Figure 5 foods-14-00256-f005:**
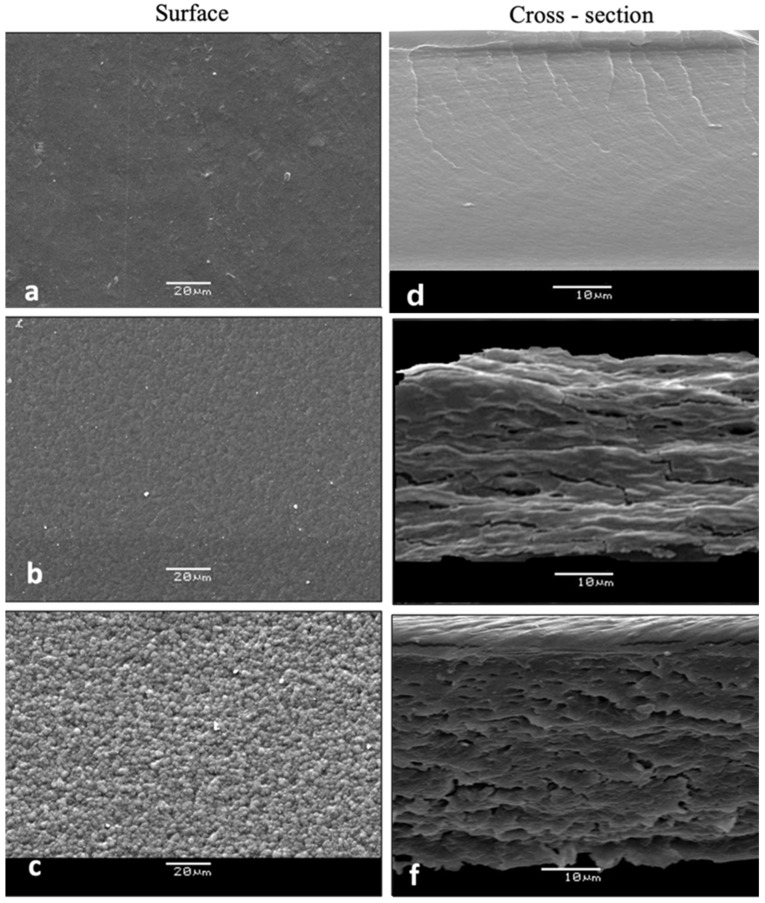
SEM micrographs showing the surface structure of (**a**) control alginate-chitosan film, and with the addition of (**b**) EOT 2%, (**c**) EOT 3%, and cross-section of (**d**) control alginate-chitosan film and with addition of (**e**) EOT 2% (**f**) EOT 3%. Oil (magnification 3000×).

**Figure 6 foods-14-00256-f006:**
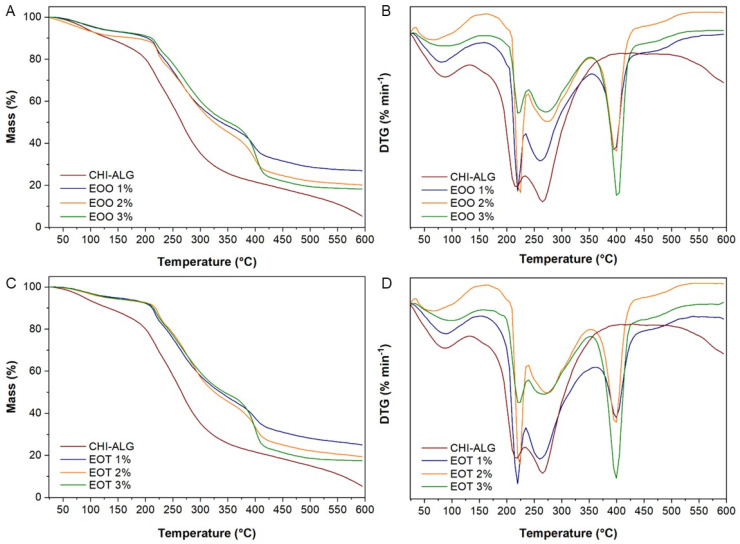
Thermogravimetric curves (TG-DTG) of chitosan (CHI)-alginate (ALG) based films loaded with different concentrations of *Origanum vulgare* (**A**,**B**) and *Thymus vulgaris* (**C**,**D**) essential oils.

**Figure 7 foods-14-00256-f007:**
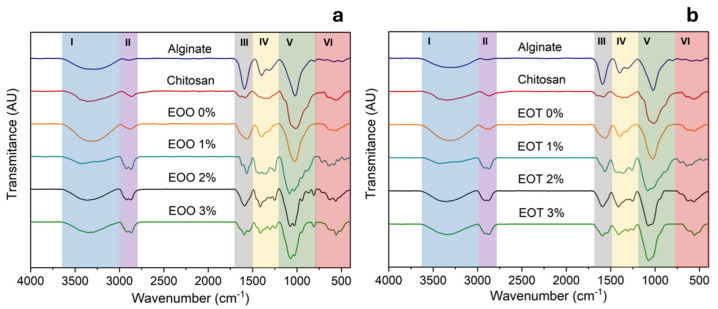
Normalized FTIR-ATR spectra for CHI-ALG, control film and films loaded with EOO% (**a**) and EOT% (**b**).

**Figure 8 foods-14-00256-f008:**
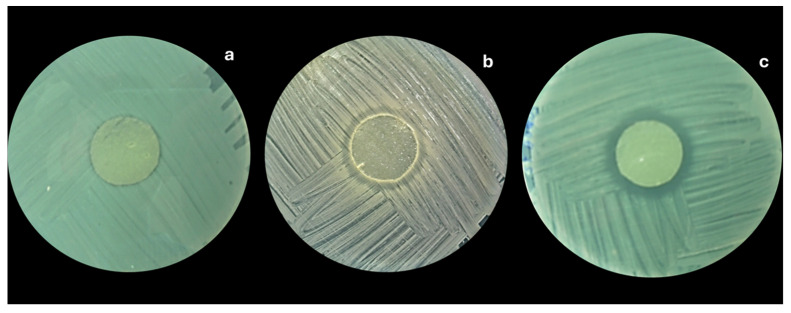
Example of the formation of inhibition halos generated by alginate-chitosan films with EOO, on *Staphylococcus aureus*: (**a**) 1%, (**b**) 2% and (**c**) 3% EOO.

**Table 1 foods-14-00256-t001:** MIC and MBC of oregano and thyme essential oils for the different strains used.

Strains	Antibacterial Parameter			
	MIC (mg/mL)		MBC (mg/mL)	
	EOT	EOO	EOT	EOO
*E. coli.*	1.39 ± 0.03 ^aA^	0.69 ± 0.03 ^bA^	2.78 ± 0.04 ^aA^	2.78 ± 0.04 ^aA^
*S. enterica*	1.39 ± 0.03 ^aA^	0.69 ± 0.03 ^bA^	5.56 ± 0.04 ^aB^	2.78 ± 0.04 ^bA^
*S. aureus*	0.69 ± 0.03 ^aB^	0.35 ± 0.03 ^bB^	2.78 ± 0.04 ^aA^	1.39 ± 0.04 ^bB^
*L. monocytogenes*	0.35 ± 0.03 ^aC^	0.17 ± 0.03 ^bC^	2.78 ± 0.04 ^aA^	1.39 ± 0.04 ^bB^

MIC: minimum inhibitory concentration; MBC: minimum bactericidal concentration; EOO: oregano oil; EOT: thyme oil. Data are reported as mean ± standard deviation (*n* = 5). Mean values with different letters indicate a significant difference between columns. Different capital letters indicate significant difference between lines (*p* < 0.05).

**Table 2 foods-14-00256-t002:** Effect of adding oregano and thyme oils in different concentrations on physical properties of alginate-chitosan films.

Essential Oil Type	%EO	Thickness (µm)	Color (CIELAB Space)						WVP (g/msPa) × 10^−9^
			L*	a*	b*	ΔE	YI	WI	
Control	0.0	34.9 ± 3.7 ^a^	98.5 ± 0.17 ^a^	−1.5 ± 0.18 ^a^	5.5 ± 0.18 ^a^	4.26 ± 0.21 ^a^	7.94 ± 0.27 ^a^	94.12 ± 0.20 ^a^	4.03 + 0.47 ^a^
EOO	1.0	38.2 ± 0.7 ^a^	88.3 ± 0.20 ^b^	−3.5 ± 0.22 ^b^	11.7 ± 0.16 ^b^	12.1 ± 0.11 ^b^	18.9 ± 0.23 ^b^	83.1 ± 0.10 ^b^	3.19 + 0.15 ^b^
	2.0	37.7 ± 1.2 ^a^	87.2 ± 0.47 ^c^	−3.6 ± 0.08 ^b^	11.6 ± 0.13 ^b^	13.0 ± 0.37 ^c^	19.1 ± 0.18 ^b^	82.4 ± 0.29 ^c^	2.47 + 1.00 ^b^
	3.0	37.9 ± 0.9 ^a^	86.4 ± 0.09 ^c^	−3.5 ± 0.15 ^b^	11.6 ± 0.07 ^b^	13.6 ± 0.06 ^c^	19.2 ± 0.11 ^b^	81.8 ± 0.05 ^c^	1.80 + 0.39 ^c^
EOT	1.0	37.8 ± 0.5 ^a^	87.8 ± 0.10 ^b^	-3.3 + 0.14 ^b^	11.5 ± 0.07 ^b^	12.3 ± 0.06 ^b^	18.7 ± 0.11 ^b^	82.9 ± 0.04 ^b^	3.04 ± 0.35 ^b^
	2.0	38.2 ± 0.8 ^a^	86.8 ± 0.04 ^c^	−3.7 ± 0.03 ^c^	11.6 ± 0.03 ^b^	13.3 ± 0.04 ^c^	19.1 ± 0.05 ^c^	82.0 ± 0.04 ^c^	2.45 ± 0.16 ^b^
	3.0	38.0 ± 0.5 ^a^	86.3 ± 0.03 ^d^	−3.4 ± 0.02 ^b^	12.0 ± 0.14 ^c^	13.9 ± 0.05 ^d^	19.7 ± 0.22 ^d^	81.5 ± 0.07 ^d^	1.65 ± 0.69 ^c^

ΔE: colour delta; YI: yellowness index; WI: whiteness index; WVP: water vapor permeability; EOO: oregano oil; EOT: thyme oil. Data are reported as mean ± standard deviation (*n* = 5). Mean values with different letters in the same column indicate a significant difference (*p* < 0.05).

**Table 3 foods-14-00256-t003:** Thermal analysis of alginate-chitosan films loaded with *Thymus vulgaris* and *Origanum vulgare* essential oil.

Sample	Temperature (°C)			Weight Loss (%)
	Onset	Peak	End	
Chitosan (CHI)	24	66	135	3.8
	251	310	402	50.4
Alginate (ALG)	25	69	185	10.6
	191	244	266	24.0
	266	284	319	10.9
	319	399	544	11.0
CHI-ALG	30	87	133	10.1
	134	216	234	24.6
	235	265	385	42.4
EOT 1%	24	90	156	6.4
	179	220	235	12.9
	236	260	360	34.9
	361	400	442	14.7
	443	477	525	4.2
EOT 2%	26	73	157	5.9
	195	225	239	11.1
	240	274	351	37.0
	352	399	443	18.8
	444	476	523	4.0
EOT 3%	28	102	156	5.9
	204	223	240	11.4
	241	267	352	41.3
	353	399	443	27.5
	444	477	524	3.6
EOO 1%	24	81	149	7.0
	205	219	235	10
	236	261	355	32.2
	356	396	438	15.5
	439	476	523	4.2
EOO 2%	24	70	145	9.2
	205	224	239	12.0
	240	273	353	32.0
	354	399	438	19.2
	439	476	523	4.0
EOO 3%	26	83	149	6.8
	204	221	241	18.3
	242	273	353	30.9
	354	400	438	26.8
	439	476	522	3.8

**Table 4 foods-14-00256-t004:** Antibacterial effects of chitosan/alginate films with essential oils of oregano and thyme.

Concentration (%)	Inhibition Zone by Type of Essential Oil (Diameter in mm)							
	*E. coli*		*S. enterica*		*S. aureus*		*L. monocytogenes*	
	EOO	EOT	EOO	EOT	EOO	EOT	EOO	EOT
0.0	0 ^Aa^	0 ^Aa^	0 ^Aa^	0 ^Aa^	0 ^Aa^	0 ^Aa^	0 ^Aa^	0 ^Aa^
1.0	26 ± 1.2 ^Ba^	30 ± 0.9 ^Bb^	22 ± 0.2 ^Ba^	26 ± 0.8 ^Ba^	24 ± 1.4 ^Ba^	27 ± 0.09 ^Ba^	27 ± 1.8 ^Ba^	40 ± 0.9 ^Bc^
2.0	25 ± 0.9 ^Ba^	39 ± 1.5 ^Cb^	28 ± 1.6 ^Ca^	28 ± 2.1 ^Ba^	30 ± 0.1 ^Ca^	38 ± 0.13 ^Cb^	37 ± 1.3 ^Cb^	46 ± 1.5 ^Cc^
3.0	39 ± 1.3 ^Ca^	41 ± 1.3 ^Ca^	35 ± 0.8 ^Da^	36 ± 0.9 ^Ca^	43 ± 1.2 ^Da^	49 ± 0.09 ^Db^	49 ± 1.3 ^Db^	55 ± 1.3 ^Db^

EOO: oregano oil; EOT: thyme oil. Capital letters different indicate significant differences between lines; small letters different indicate significant differences between columns (*p* < 0.05).

## Data Availability

The original contributions presented in the study are included in the article, further inquiries can be directed to the corresponding author.

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
