# Peer review of "Development of Alginate-Chitosan Bioactive Films Containing Essential Oils for Use in Food Packaging"

_foods, 2025, doi:10.3390/foods14020256_

Round 1
Reviewer 1 Report
Comments and Suggestions for Authors
1. The author should address the current research's novelty in the introduction's last paragraph. There is no explanation of the advantages of the recent work on the published work in the introduction.
2. The sentence, " In this line, this study focuses on investigating and improving the interactions between alginate, chitosan, and essential oils to enhance the physical, mechanical, and antibacterial properties of bioactive films applicable in food packaging." should be rephrased.
3. Kindly add the full details of the used materials.
4. Many typo-error and grammatical errors are found in the whole manuscript; this type of error should be fixed.
5. The author should add the FTIR data of all samples, and the interaction of the alginate and chitosan should be addressed in the results and discussion. The functional group interaction of the oils with the alginate/chitosan should be highlighted with the help of the FTIR data.
6. The author should add the stress-strain curve of the samples.
7. In Figure 5, kindly add the inhibition film of all concentration samples.
8. The conclusion does not look attractive; some quantitative results should be added.
Author Response
"Please see the attachment."

Reviewer 2 Report
Comments and Suggestions for Authors
The application of antibacterial properties of essential oils and the development of active films for food are both research directions of concern in the food industry. This manuscript explored the development of bioactive films using oregano and thyme essential oils combined with alginate and chitosan. The study has successfully produced a composite film with high elasticity and strong antimicrobial properties, showing promising potential for food packaging applications. The manuscript is well-written, with appropriate structure, comprehensive research content, accurate result analysis, and thorough discussion. Here are some comments:
1. Adjust the hardness unit in Table 2 to be consistent with the analysis section (mm or μm).
2. According to Table 2, essential oils did not significantly affect film thickness; therefore, the statement "the incorporation of essential oil increased the thickness of the films" should be removed from the abstract.
3. Why does the abstract only emphasize significant changes in b* when L*, a*, b*, ΔE, YI, and WI all show significant differences compared to the control?
4. Standardize the scale bars between Figure 3d and 3ef.
5. SEM micrographs for EOT treatment should be provided.
6. Revise Table 4 footnotes to clarify whether different or same letters indicate significant differences. The significance test results in Table 4 need to be re-examined.
Author Response
"Please see the attachment."

Reviewer 3 Report
Comments and Suggestions for Authors
The abstract should be rewritten; please add some results.
Keyword „edible films” should be added
The introduction should be improved with more general information about edible films. I recommend starting with this topic, then biopolymers and essential oils. The same for section 2.1.
The drying process of film-forming solutions is not clear; why is there such a big range of time?
Use dots for values
The film was conditioned before mechanical properties analyses, but what about others?
Section 2.2.6. provide magnification, for Fig. 3 too
Conclusions should be completed with more observations
Add digital pictures of the films
Add practical applications for these films or at least the potential ones
Author Response
"Please see the attachment."

Reviewer 4 Report
Comments and Suggestions for Authors
The paper “Development of bioactive films for use in food: Analysis of the impact of ionic interaction between alginate, chitosan and essential oils on their physical and antibacterial properties” is a standard paper describing the preparation of bioactive films based on alginate and chitosan with variable content (1 %, 2 % and 3 %) of essential oils (EOs) extracted from oregano (Origanum vulgare) and thyme (Thymus vulgaris) plants to be used as food packaging. The manuscript is well written (with some exceptions, see below), and the results are credible. The investigation methods are rather classic; therefore the degree of novelty is relatively low. The major lack is a comprehensive comparison with many other films (only me, I review more than 15 paper on this subject in the past several years) reported in the literature and to compare produced films in this study with the best on market/literature! The films are compared among them and just a “shy” offer (alginate/chitosan-3.0% EOT) was proposed as a “good alternative” (the bed alternative is not specified). My real problem is related to the title of the manuscript. “Analysis of the impact of ionic interaction…”. The term “ionic” is used 3 times: once in title, once in Introduction, and once in Bibliography. In my opinion the paper does not present any analysis of ionic interactions. Indeed there are some discussions which may be associated to the subject but since the concept is found in title this should be the main focus, which is not. Nevertheless, since I did not detect major flaws, my recommendation for editor was that the paper could be some minor considerations. Some of my other observations are listed below: 1. Title: “Development of bioactive films for use in food:” – Please clarify in the title the fact that these films are not used in food, per se, but as food packaging! 2. Keywords. In my opinion just the separate “ingredients” cannot give a complete view of the main materils, e.g. bioactive films used in food packaging! Please add also this keyword. 3. Abstract: “Regarding antibacterial capacity, as the concentration of essential oil increases, the antibacterial capacity also increases.” – Please rephrase. Be more direct! 4. Introduction “However, the chitosan application in the food packaging industry has limited oxygen barrier properties and is sensitive to environmental humidity [20].” Unclear! Please revise the sentence! 5. 2.1 Materials “The essential oils used in this study were oregano (Origanum vulgare) and thyme (Thy-mus vulgaris) obtained by steam distillation from company Sigma-Aldrich (St. Louis, MO, USA)” Please revise the phrase to result that oregano and thyme are plant and not EO! 6. Idem “To determine the antibacterial characteristics of different essential oils, the follow-ing bacterial cultures Escherichia coli (ATCC 25922), Salmonella enteritidis (ATCC 13076), Staphylococcus aureus (ATCC 6538), and Listeria monocytogenes (ATCC 7644) obtained from the culture collection of the Food Microbiology Laboratory of the Department of Food Engineering of the Universidad del Bio Bio, Chile.” A verb is missing! 7. Section 2.2.7 “thermal” to “temperature”. The physical parameter is temperature. “Thermal” is just related to! 8. Section 3.2.2 “Film color can negatively affect consumer acceptability.” Be more explicit. “Film color” is a property. Not the property, but the “value” of that property can produce effects. E.g. “a yellow hue”. 9. “Our results indicated that both oregano and thyme essential oils significantly reduced the luminosity of the films.” “Luminosity” is a characteristic of a light source. Are your films producing light? If not, please change accordingly (e.g. reflectance, transparence, transmission index, etc)! 10. Figure 1 and 2. Please use also minor ticks to increase the precision of reading the measured values! 11. Section 3.2.6. Thermogravimetry analysis. These measurements were performed up to 600 C, a temperature much higher compared to those of regular use. The results seems interesting but to make sense please associate your findings to other properties. My expectations were related to a discussion function of essential oil concentration! 12. Section 3.2.7. Antibacterial effect of alginate-chitosan films! “On average, the increase from 1% to 3% of EOO increased the antimicrobial capacity against Gram-negative bacteria from 24 to 37 mm (154% increase), while for EOT, it was from 28 to 39 mm (139% increase).” Please name the quantity measured in mm! (E.g. from 24 to 37 mm and from 28 to 39 mm). Justify your choice. Why diameter! Why not the area? Why not the thickness? The diameter of film may affect your percentage of increase. Please comment of this aspect! 13. Figure 5. Please provide in figure legend all necessary information!
Author Response
"Please see the attachment."

Reviewer 5 Report
Comments and Suggestions for Authors
Dear authors, here are my suggestions for improving the manuscript:
Initially, line numbers are missing. Authors removed by mistake? This makes it difficult to indicate more specific corrections in the work
Title: why the tile mention “Analysis of the impact of ionic interaction”? There is no deep discussion of what this ionic interaction consists of throughout the work.
Abstract: please include more specific data.
Introduction: Please clarify the novelty of this study compared to existing work.
MM:
- Statistical analyses: replicates were performed?
- How about the multivariate analyses to evaluate the interactions between variables? E.g., oil type, concentration, and film properties.
- How authors evaluated the interaction between chitosan and alginate? Films were analyzed individually to compare?
- The drying time (48-62 hours) may influence reproducibility and scalability? This variability in drying times should be statistically examined or justified.
- Concentration of EOs tested just evaluated 1-3% concentrations. Why? Authors realized previous studies to obtain these concentrations? And concentrations below and above this range?
- Authors did not considered to evaluate long-term efficacy? How about the stability of the film under different food storage conditions?
- Considering that authors only evaluated microbial effects, why so limited bacterial strains were utilized (only 4)? Spoilage fungi and other pathogens are missing.
- Finally, to correctly evaluate bioactive properties in films, the determination of the bioactive compounds of the oils is crucial. This would allow a specific conclusion about the results obtained for the effects on the microorganisms studied (main objective of the work).
Results and discussion:
- Please provide stronger evidences for long-term food safety benefits and discuss practical implications (e.g., industrial scalability, cost-effectiveness). The work aims to evaluate a film for food, but there is little discussion with previous studies with biofilms and food.
- Please clarify if inhibition zones correspond directly to in vitro food safety improvements.
- Some tables lack clarity in units and standard deviations.
- How about the economic feasibility and production scalability?
- The discussion should better contextualize results highlighting innovations or confirming prior findings.
- The SEM images and thermal analysis suggest structural changes due to EOs, but their practical implications (e.g., shelf-life extension, compatibility with different food types) are not discussed.
- And the films’ sensory impact on food products (e.g., odor or flavor impartation due to EOs)?
-There are discrepancies between some descriptions and data in Table 2. Should be interesting to include in figure 3 the magnification in the legend. In addition, please include a more detailed legend for figure 5.
Conclusion: Lacks detailed implications or suggestions for future studies. It does not sufficiently emphasize the broader implications, such as potential applications in extending food shelf life and improving sustainability. The conclusion needs significant improvement, the text was limited to summarizing the results of the work. In addition, in the opinion of this reviewer and the review realized, the authors' statement "Therefore, this film can be a good alternative for bioactive packaging applications." cannot be substantiated based on the study carried out.
Author Response
"Please see the attachment."

Round 2
Reviewer 3 Report
Comments and Suggestions for Authors
Publication has been improved and can be published in the corrected version.
Author Response
Thank you very much, dear reviewer, for your comments, advice and suggestions for improving this article.
Reviewer 5 Report
Comments and Suggestions for Authors
The authors responded to my suggestions and they have addressed all the comments appropriately. In my opinion, the manuscript is now ready for acceptance.
Author Response

(The authors gave the same response as above.)
